# Cognitive Dysfunction in Myalgic Encephalomyelitis/Chronic Fatigue Syndrome—Aetiology and Potential Treatments

**DOI:** 10.3390/ijms26051896

**Published:** 2025-02-22

**Authors:** Amolak Singh Bansal, Katharine A. Seton, Jonathan C. W. Brooks, Simon R. Carding

**Affiliations:** 1Spire Bushey Hospital, Bushey WD23 1RD, UK; asbansal1000@yahoo.com; 2Food, Microbiome and Health Research Programme, Quadram Institute Bioscience, Norwich NR4 7UQ, UK; katharine.seton@quadram.ac.uk; 3Department of Psychology, University of Liverpool, Liverpool L69 7ZA, UK; jon.brooks@liverpool.ac.uk; 4Norwich Medical School, University East Anglia, Norwich NR4 7TJ, UK

**Keywords:** myalgic encephalomyelitis, chronic fatigue syndrome, cognitive dysfunction, infection, inflammation, cytokines, immune dysfunction, cerebral blood flow, brain imaging

## Abstract

Systemic infection and inflammation impair mental function through a combination of altered attention and cognition. Here, we comprehensively review the relevant literature and report personal clinical observations to discuss the relationship between infection, peripheral inflammation, and cerebral and cognitive dysfunction in patients with myalgic encephalomyelitis/chronic fatigue syndrome (ME/CFS). Cognitive dysfunction in ME/CFS could result from low-grade persistent inflammation associated with raised pro-inflammatory cytokines. This may be caused by both infectious and non-infectious stimuli and lead to altered regional cerebral blood flow accompanied by disturbed neuronal function. Immune dysregulation that manifests as a subtle immunodeficiency or the autoimmunity targeting of one or more neuronal receptors may also be a contributing factor. Efforts to reduce low-grade systemic inflammation and viral reactivation and to improve mitochondrial energy generation in ME/CFS have the potential to improve cognitive dysfunction in this highly disabling condition.

## 1. Introduction

Myalgic encephalomyelitis/chronic fatigue syndrome (ME/CFS) is a disabling condition characterised by several symptoms including fatigability and delayed malaise after exertion, pain, sleep disturbances, headaches, sore throats, and flu-like symptoms [1,2,3,4]. Also common are impaired memory and concentration (often described by patients as brain fog), muscle weakness, disequilibrium, and gastrointestinal disturbances [4,5,6,7,8]. Collectively, these symptoms make it very difficult for patients to work or learn efficiently and for any length of time.

Self-reported cognitive impairment in the form of impaired attention, poor memory, and concentration difficulties are commonly described by ME/CFS patients, with memory and concentration problems reported by 89% of ME/CFS patients [7]. As cognitive impairment affects both occupational and social activities, it is one of the more debilitating symptoms experienced by ME/CFS patients. To alleviate cognitive symptoms, we first must understand the cognitive domains affected and the mechanisms causing these deficits.

Patients with ME/CFS often report the onset of their illness after a viral or other infection. Low grade inflammation reflected by elevated blood levels of C-reactive protein (CRP) and pro-inflammatory cytokines such as interleukin (IL)-6 and tumour necrosis factor (TNF)α are evident in many patients with ME/CFS [9,10]. Raanes and colleagues [11] have correlated psychological variables and sleep disturbance with peripheral cytokine levels, identifying associations between executive function and IL-1 and IL-6, interpersonal function and IL-6 and TNFα, and sleep and IL-1, IL-2, IL-6, and TNF alpha. Unsurprisingly, cerebral blood flow and blood–brain barrier dysfunction are more frequent in the presence of inflammation, even when this originates outside the CNS [12]. Microglial alteration of blood–brain barrier integrity is also affected by peripheral inflammation [13]. While the exact mechanism of cognitive dysfunction in ME/CFS is unclear, it is most likely multifactorial, involving several life factors that modulate neuroinflammation, neuronal dysfunction, and altered cerebral blood flow (Figure 1). Immune dysfunction, immunodeficiency, and autoimmunity directed at neuronal proteins may also contribute in at least some ME/CFS patients [14,15]. In this review, we discuss the interaction of these variables in the causation of cognitive dysfunction (CD) in ME/CFS.

## 2. Cognitive Dysfunction in ME/CFS

Cognitive function in ME/CFS has been extensively examined over the past 40 years using both subjective questionnaires and objective neurocognitive assessments [16,17,18]. Subjectively reported cognitive difficulties in memory, attention, and information processing are significantly worse in ME/CFS patients compared to healthy controls [19,20,21,22,23,24]. Impairments in these cognitive domains have been confirmed using neurocognitive assessments, although difficulties in working memory and information processing are suggested to only emerge when undertaking time-dependent tasks [25,26,27,28,29,30,31]. However, results are often inconsistent, with some studies finding no differences between ME/CFS patients and controls [19,21,22,32].

Attempts to consolidate and delineate the cognitive profile of ME/CFS have come from two systematic reviews and meta-analyses. The first reviewed 50 studies carried out between 1988 and 2008 identifying deficits in the domains of attention, memory, and reaction time [17]. The second reviewed 33 studies published between 1988 and 2019. This excluded studies with inconsistent methodologies and standards and identified significant impairments in visuo-spatial, short-term, and especially immediate memory, with impacts on working memory being consistent with a massive executive deficit in patients [16]. These meta-analyses highlighted potential sources of conflict within study findings and significant moderate to large impairments were seen in some with cognitive tests not falling under the same cognitive domain.

In addition, the choice of ME/CFS case definition likely contributed to the large heterogeneity in scores on neurocognitive assessments. Unlike case definitions published after 2000, the presence of neurological/cognitive symptoms is not required for a diagnosis of ME/CFS according to the 1994 Fukuda criteria [2,33], which is the definition used in most older cognitive function studies [18]. Indeed, more than 25% of patients diagnosed according to the 1994 Fukuda criteria report no difficulties with memory and concentration in their daily life [34]. One study assessing the proportion of participants who were impaired in each cognitive domain found impaired attention and impaired motor functioning in approximately 50% of patients, impaired executive functioning and information processing speed in 40%, impaired visual memory in 30%, and impaired verbal memory and problem -solving in under 20% of patients [35]. This suggests that different phenotypes of cognitive function impairments exist. There is therefore a need to delineate clinical subtypes for cognition-based evaluations in ME/CFS patients.

The advantage of stratification is exemplified by the findings of DeLuca et al. [36], who divided their patients into two groups according to the onset of symptoms. While people with ME/CFS showed a significant reduction in their information processing ability relative to controls, impairment in memory was more severe in those with a sudden, and presumed, infectious onset. However, the severity of the CD appears to be no different in patients with a short or long duration of illness [37].

The complexity and heterogenous symptomology of ME/CFS remains a challenge. It should, therefore, be borne in mind that other variables may affect performance, notably fatigue and its impact on a patient’s functional capacity [38,39], which historically has not been possible to measure objectively and reliably. The recent development of a patient-informed questionnaire, FUNCAP, that assesses functional capacity across eight different domains and types of daily activity, should prove useful in addressing this deficiency and providing a means of accurately assessing fatigue and its consequences in both diagnostic and research settings [40].

Chronic pain is a common complaint in ME/CFS [41]. Pain has an “attentional cost” which affects cognitive domains linked to attention [42]. However, there is evidence that the concomitant self-reported pain in ME/CFS does not appear to influence cognitive performance [21]. Other ME/CFS symptoms such as sleep disturbances could have an “attentional cost” but this link in ME/CFS has been neglected to date.

Depression and anxiety can affect cognitive functions which rely on attention [43]. An increased frequency of current instances of and a lifetime prevalence of depression and anxiety is evident in patients with ME/CFS [44,45]. However, after allowing for mood, these patients still had marked impairment of attention [44]. Furthermore, Constant et al. [26] noted the impaired cognitive function in ME/CFS to be related to impaired attention and memory, and with only a weak correlation with depression. It was also associated with a reduced reaction time but without a significant impairment in fine motor speed, vocabulary, reasoning, and global functioning [17].

## 3. Cognition and Neurological Pathways

The ascending reticular activating system (ARAS), comprising a network of neurones in the brainstem, has long been considered important in the arousal mechanisms that underpin attention and was first elucidated in positron emission studies [46]. The important role of the posterior intralaminar thalamic nuclei has been highlighted by Quiroz-Padilla et al. [47], particularly in relation to attention, learning, and memory. As discussed in more detail in the altered brain function section below, different MRI modalities have described structural changes in specific regions of the brain. These include ischaemia of the thalamus and midbrain dysfunction associated with impaired concentration [47], a decrease in midbrain white matter volume [48], and absent connections between the medulla and midbrain nuclei [49]. During tasks, ”connectivity deficits” were seen within the brainstem between the medulla and cuneiform nucleus and between the brainstem, the hippocampus, and the intralaminar thalamus. Furthermore, increased symptom severity in ME/CFS was associated with weaker connectivity between some ARAS nuclei. However, no such changes were observed in adolescents with ME/CFS, even though they consistently displayed higher subjective fatigue [50], and the controls outperformed them overall on the measures of processing speed, continued attention, and new learning.

## 4. Structural Brain Abnormalities

Analysis of structural MRI data with voxel-based morphometry (VBM) has shown a global reduction in grey matter in patients with ME/CFS compared to healthy age-matched controls [51]. Interestingly, a significant positive linear relationship between daily activity and total grey matter volume was observed in patients alone. However, Perrin et al. [52] reported “no abnormal patterns in rate and extent of brain atrophy, ventricle volume, white matter lesions, cerebral blood flow or aqueductal CSF flow” in 18 patients with CFS compared to nine healthy controls. More recently, ultra-high field MRI at 7 tesla has been used to compare the volume of brainstem structures between patients with ME/CFS, people with long COVID, and healthy controls [53]. Patients with ME/CFS and those with long COVID exhibited increased volumes in brainstem structures compared to controls. However, it is not clear whether the estimated subcortical volumes were adjusted for differences in total intracranial volume (TIV)—a recommended normalisation step for such studies [54]. Notably, the study by Finkelmeyer et al. [55] addressed this by accounting for TIV in their analyses. They found that regions associated with interoceptive awareness (insular cortex) and stress (amygdala) showed increased volume, and the brainstem decreased in volume when compared to controls.

## 5. Altered Brain Function

The impact of ME/CFS on brain function can be assessed by functional MRI [56] which uses Blood Oxygen Level-Dependent (BOLD) contrast to infer changes in brain activity. This is associated with regional alterations in cerebral blood volume (CBV) and cerebral blood flow (CBF) and is an indirect measurement of the brain’s electrical activity [57]. Mentally fatigued ME/CFS patients elicited brain activity more widely than healthy controls in areas involved in working memory [58] and in medial prefrontal regions including the cingulate gyrus [59]. More recently, Baraniuk et al. [60] showed that when performing a verbal working memory task (n-back), patients with ME/CFS demonstrated a significant increase in activity within periaqueductal grey, dorsal, and median raphe, right midbrain reticular formation, the parabrachial complex, and the locus coeruleus when compared to controls. This pattern of increased activity post-exercise was also observed within the medial prefrontal cortex (here, part of the default mode network—DMN) [61] when studied with resting-state fMRI [62]. Similarly, Manca et al. [63] utilised resting-state fMRI to reveal increased connectivity between the right insular cortex (part of the salience network—SN) [64] and the orbitofrontal cortex in patients compared to controls. Not all brain imaging studies are, however, consistent in their findings, with some demonstrating reduced activation in the dorsolateral prefrontal and parietal cortices or the superior occipital/posterior parietal lobe of patients [59,65]. Lee et al. [66], in their recent metanalysis, noted reduced activity in the insula and thalamus and suggested that disrupted connections in the limbic system contributed to symptomatology. Considering the indirect nature of neurovascular coupling that gives rise to BOLD contrast [67], it is worth noting that a global reduction in CBF has been reported in ME/CFS patients when compared to healthy controls [68]. Such a reduction would have implications for measured BOLD signals and could lead to apparent reductions in activity when compared to healthy controls.

Whilst fMRI studies fail to produce consistent findings, there is good reason to believe that altered brainstem function could underlie some of the symptoms experienced. One of the earliest neuroimaging findings in ME/CFS was of hypoperfusion of the brainstem, measured with single photon emission computed tomography (SPECT) [69]. Importantly, brainstem grey matter volume shows a strong correlation with pulse pressure in patients with ME/CFS [48]. This has raised the possibility of impaired cerebrovascular autoregulation, particularly as there were associated changes in the white matter volume of the deep prefrontal area, the caudal basal pons, and the hypothalamus [48]. Any consequent impairment of ARAS activity in the brainstem would likely have an adverse effect on attention and arousal mechanisms, leading to memory dysfunction. Enhanced functional connectivity between the brainstem and forebrain structures [70] has been observed in ME/CFS patients and was positively correlated with fatigue produced by the continuous performance of a demanding cognitive task. Connectivity between brainstem nuclei associated with autonomic regulation and ARAS, e.g., the rostral ventrolateral medulla and the nucleus cuneiformis, have also been shown to be altered in ME/CFS and associated with symptom severity [49].

## 6. Neuroinflammation

The contribution that neuroinflammation makes to CNS pathologies associated with ME/CFS comes from positron emission tomography (PET) brain imaging using a radioligand for the translocator protein (TSPO), a protein belonging to the mitochondrial permeability transition pore (MPTP) complex and produced by activated microglia [71]. Using this approach, Nakatomi and colleagues revealed a correlation between cognitive impairment in ME/CFS patients and PET-signals, especially in a region between the mid-pons and thalamus [23] consistent with the hypothesis of neuroinflammation in ME/CFS. However, no such difference in TSPO binding between ME/CFS patients and controls was found in a more recent study [72]. This may in part be due to the allelic dependence of ligand affinity, leading to variations in TSPO binding between subjects. Indeed, the interpretation of TSPO binding is complex, with the authors of a recent review stating: “At present, it is not clear whether the presence of TSPO PET signal indicates a predominantly pro-inflammatory or anti-inflammatory state, nor whether the signal indicates the presence of destructive or reparative cell states.” [73]. The development of newer radioligands may help address these shortcomings and provide definitive evidence of neuroinflammation in ME/CFS.

The last technique that provides a means to assess neuroinflammation non-invasively is magnetic resonance spectroscopy (MRS). This approach has previously been used to demonstrate reductions in N-acetylaspartate (NAA), a putative neuronal marker in the hippocampus of patients with ME/CFS [74]. More recently, using a spectroscopic imaging approach, widespread alterations in the ratios (to Creatine) of Choline, myo-Inositol, and Lactate were found across the brains of patients with ME/CFS [75]. Concentrations of these metabolites are thought to be elevated by neuroinflammation. The same methodology was also used to infer brain temperature, which has been suggested as a proxy measure for neuroinflammation. Compared to controls, significantly elevated brain temperature was observed in the insular and frontal cortices, the thalamus, the putamen (all right hemisphere), and the cerebellum.

The origins and sources of inflammatory mediators promoting CNS pathology noted in brain imaging studies of ME/CFS patients are presently unclear.

## 7. Viral Infections and Cognitive Dysfunction

A high proportion of ME/CFS patients report the onset of their illness after a viral illness [76]. Elevated levels of certain proteins in peripheral blood mononuclear cells (PMBCs) or cytokines in the sera of ME/CFS patients important in controlling viral replication are suggestive of chronic viral infection [77,78,79]. Among the viruses commonly associated with ME/CFS are human herpes virus (HHV) 6 and Epstein–Barr virus (EBV) [80]. However, the detection of viruses in ME/CFS patients is inconsistent with both serological and molecular-based assays, not in frequently producing conflicting results [81,82,83,84].

A subset of patients develop ME/CFS following EBV infection and report long-term impaired memory, concentration, headaches, and other symptoms [85]. Interestingly, chronic EBV infection may also be associated with CD. EBV-encoded deoxyuridine triphosphate nucleotidohydrolase (dUTPase) can stimulate anxiety and sickness behaviour in female mice with an altered expression of multiple proteins and genes important in blood–brain barrier integrity, fatigue, pain, and synapse structure [86]. Furthermore, a subset of ME/CFS patients demonstrated prolonged and significantly elevated serum-neutralising antibodies against this protein that inversely correlated with their symptoms [87]. Finally, Kogelnik et al. [88] observed a significant improvement in the physical and mental functioning of 9/12 ME/CFS patients with high titre EBV and HHV-6 IgG antibodies receiving 6 months of valgancyclovir in an open-label study. Other specific viral infections that can lead to impaired memory and ME/CFS-like symptomology include Nipah and parvovirus B19, which can precede ME/CFS [89].

Retroviruses and SARS-CoV-2 infection can, in some acutely infected individuals, lead to the development of prolonged ill health and post-COVID-19 syndrome referred to as long COVID (or post-acute sequelae of COVID (PASC)). The latter comprises several disabling symptoms persisting 12 weeks after a confirmed SARS-CoV-2 infection [90] and long COVID shares several overlapping symptoms with ME/CFS. These include prolonged fatigue, cognitive impairment/brain fog, post-exertional malaise, compromised short-term memory, unrefreshing sleep, musculoskeletal pain, and headaches [90,91]. It is therefore not surprising that a significant number of long COVID patients have been diagnosed with ME/CFS [92,93]. CD, memory loss, and attention disorder are also common in those with long COVID [94], affecting between one-third and one-quarter of patients [95,96]. The cause of the CD in long-COVID is unclear and likely multifactorial, involving a combination of one or more mechanisms [97]. Among these are viral persistence and neuroinvasion [98], immune dysfunction and subtle autoimmunity leading to changes in neuronal signalling [99], changes in endothelial function affecting cerebral perfusion [100], microvascular clot formation and complement activation [101], and altered neurotransmitter metabolism [102]. It is also possible that the immune dysfunction following SARS-CoV-2 infection permits the reactivation of EBV [103] and HHV-6, with all the viruses being collectively responsible either directly or via autoimmune mechanisms for the CD [104].

Human endogenous retroviruses (HERVs) and their aberrant reactivation are another source of “retroviral infection” that can lead to infection-like symptoms, such as pain, fatigue, and the immune and metabolic disturbances commonly seen in ME/CFS [105,106]. HERV activation in the lymphocytes of COVID-19 patients has been correlated with inflammatory markers and pneumonia severity [107], and in multiple sclerosis (MS) and amyotrophic lateral sclerosis, it can lead to demyelination, inflammation, and cognitive problems [108]. HERV expression, principally the K and W families, has been described in the PBMCs of ME/CFS patients in some but not all studies [109,110,111]. HERV expression has also been detected in duodenal tissue samples from 8/12 patients with ME/CFS and in all of those with gastritis with HERV-*gag*, *pol* colocalising with CD303^+^, CD86^+^, and MHC II^+^ immune (dendritic) cells [112]. Of note, EBV and anti-viral IFN alpha (IFNα) can transactivate HERVs (K18), providing a link between herpesvirus infection and the reactivation of HERVs which encode molecules (superantigens) with the ability to interfere with or modulate immune and T cell function in their host organism [113,114]. This could help explain disease aetiology in some infection-related ME/CFS patients.

## 8. Mechanisms Linking Infection, Inflammation, and Cognitive Dysfunction

The exact mechanism of fatigue and cognitive and memory dysfunction associated with, and following, inflammation and infection is unclear, and the important variables are detailed in Table 1. However, the administration of IL-6 in amounts equivalent to those seen in acute infection has been shown to produce several changes, both peripherally and centrally [115]. These include increased CRP, ACTH, and cortisol but reduced thyroid stimulating hormone (TSH). Subjects reported fatigue and felt more inactive and less capable of concentrating than after a placebo. Their sleep architecture also changed, and slow-wave sleep decreased during the first half and increased during the second half of sleep. Rapid eye movement sleep during the entire nocturnal sleep time was also significantly decreased. Conversely, sleep disturbances because of narcolepsy, sleep apnoea, and idiopathic hypersomnia have been associated with increased levels of IL-6 and TNFα [116]. In terms of immune dysfunction, disrupted sleep has been shown to promote both central and peripheral inflammation [117] and reduce the secretion of IL-7, which is critical to T cell memory [118] and preventing recurrent infection. This has relevance for patients with ME/CFS who have non-restorative sleep that is associated with a combination of delayed initiation and maintenance, as well as early morning wakening in some with prolonged sleeping-in with others.

From an immunological perspective, neuropsychological stress leads to the activation of the hypothalamic–pituitary–adrenal (HPA) axis via intermediary IL-1. The subsequent glucocorticosteroid secretion, if not contained, can lead to problems with memory and attention as well as impaired neurocognition in the long term [119]. How peripheral viral infections leads to impaired mood accompanied by altered memory and attention is partly suggested by the flattening of the ACTH and cortisol slope produced by IFNα/ribaviran in patients with Hepatitis C virus (HCV) infection [120]. While this flattening correlated with measures of depression and fatigue, there was no clear correlation with increases in TNFα and soluble TNFα receptor 2.

Toll-like receptors (TLRs) and other microbial pattern recognition receptors (PRRs) produce inflammatory cytokines when stimulated by bacterial lipopolysaccharide (LPS), as well as bacterial and viral RNA and DNA [121]. Several PRRs and TLRs have been detected in different parts of the CNS, contributing to processes as distinct as pain modulation [122] and memory function [123]. Thus, peripheral infection accompanied by the release of endotoxins and/or DNA/RNA may stimulate inflammation mediated neural dysfunction within specific parts of the CNS as these cross the blood–brain barrier. TLR stimulation may either directly, or via the release of pro-inflammatory and antiviral cytokines also lead to memory dysfunction as well as many of the other symptoms seen in ME/CFS.

Mechanistically, peripherally synthesised pro-inflammatory cytokines may lead to depression and altered memory and attention through a disturbed synthesis of certain neurotransmitters. This is often seen in HCV infection and certain cancers treated with IFNα [124,125]. Thus, IL-1β, interferon (IFN) α, IFNγ, and TNFα can affect 5-hydroxytryptamine (HT) metabolism directly and/or indirectly by stimulating the enzyme indoleamine 2,3-dioxygenase [126]. This may then lead to the depletion of peripheral tryptophan which is needed for the synthesis of several neurotransmitters, including serotonin.

Descriptions of deficits in the cellular immune system in ME/CFS [127] raise the possibility that these immune defects may predispose ME/CFS patients to viral infections or the reactivation of previously acquired viral infections. However, Cameron et al. [128] have discounted the importance of EBV, CMV, and HHV6 based on serological and molecular analyses showing the absence of significant viral antibodies and gene copy numbers in ME/CFS patient samples. In this case, active viral infection may not be necessary. Thus, the presence of certain viral proteins such as the EBV/HHV-6A dUTPase has been shown to inhibit the replication of human PBMCs in vitro and to increase the production of TNFα, IL1β, IL6, IL8, and the immunoregulatory IL10 [129,130]. Additionally, it can elicit the production of secondary immunoregulatory proteins such as activin A, which have potent immune regulator functionality [131]. Mechanistically, this has involved the binding of TLR 2 and the subsequent activation of nuclear factor (NF)-kappa B through the recruitment of the MyD88 adaptor molecule [132]. Additionally, HHV6 reactivation has been shown to be capable of inducing mitochondrial fragmentation and impairing ATP synthesis [133] which would clearly reduce neural activity mediating cognition. Moreover, poly I-C (polyinosinic-polycytidylic acid, a synthetic analogue of double-stranded RNA) injected peripherally can induce immune changes in the cortex, hippocampus, and hypothalamus [134]. Furthermore, HERV-K18-encoded superantigens expressed in response to EBV infection and IFNα can modulate T and B cell differentiation, antibody production, and NK cell function. This provides another link between virus infections and the activation of endogenous superantigens leading to chronic immune activation, inflammation, and possibly autoimmunity in susceptible individuals [135].

As previously discussed, overt or subclinical infection can, via peripherally generated pro-inflammatory cytokines, contribute to a worsening of fatigue, cognition, and memory in those with ME/CFS. A sensation of feeling unwell with the aching of muscles and knowledge that movement can unpredictably exacerbate symptoms then leads to reduced muscle usage. As functioning muscle tissue is one of the few sources of brain-derived neurotrophic factor, this leads to impaired memory and cognition [136]. Additionally, viral DNA/RNA may directly stimulate one or more TLRs on microglial or hippocampus cells, leading to the production of pro-inflammatory cytokines. These may then alter cerebral blood flow, particularly to the hippocampus and ARAS, leading to disturbed memory. The situation is likely to be worse in those with joint hypermobility syndrome (JHS) and postural orthostatic tachycardia syndrome (POTS), which is not infrequent in ME/CFS, and who have impaired regulation of blood pressure and heart rate and significant CD after changes in posture [137]. Interestingly, variably subtle cardiovascular and blood clotting functions, similar to those seen in post-acute sequelae SARS-CoV-2 infection, may also be evident in ME/CFS and may impair both central and peripheral oxygen and nutrient supply [138]. Other factors that may occasionally adversely impact cerebral perfusion include subtle cerebral vasculitis [139,140] and possibly variants of anti-neutrophil cytoplasmic auto-antibodies [141]. There is also the possibility that virus and stress-induced T and B cell memory dysfunction [142] may promote autoimmunity to nuclear proteins and neuronal receptors [143]. At a speculative level, this may produce many of the subtle cognitive changes seen in ME/CFS.

## 9. Treatment of CD

ME/CFS has symptoms and sometimes dysfunctional immune pathways that overlap with several other conditions, including as previously mentioned, PASC, but also several of the systemic connective tissue disorders [144] and also neurological conditions, such as MS [145] and Parkinson’s disease. The CD seen in ME/CFS is also evident in those with depression and anxiety [146]. Unsurprisingly, early non-pharmacological therapeutic interventions for ME/CFS have often included participants with conditions causing CFS-like symptoms, owing to the use of less stringent criteria [147]. Additionally, several have not dealt adequately with missing outcome data, inadequate randomisation, and bias in the reporting of outcome data, and have had issues with uniform implementation or changes in the intervention plan [148]. As may be expected, the inclusion of populations with chronic fatigue arising from conditions such as depression, anxiety, and medically undiagnosed conditions has complicated the interpretation of results. This has contributed to a heated debate about the efficacy of treatments such as cognitive behavioural therapy (CBT) and graded exercise therapy (GET), used primarily in the PACE trial [149], for those meeting stricter criteria for ME/CFS, such as the Canadian Consensus Criteria or the International Consensus Criteria. Importantly, few interventions have been rigorous enough to provide unequivocal economic benefit despite the huge financial cost of ME/CFS to the economy [148]. Notwithstanding, CBT and possibly GET were considered beneficial by Kim et al. [150] but often at huge financial costs and without definitive improvement in CD. Other therapies that they reported to be statistically beneficial included Staphypan Berna, Poly(I):poly(C_12_U), and a combination of Coenzyme Q_10_ (CoQ_10_) with Nicotine Adenine Dinucleotide H [150], although the numbers investigated were small.

We have recently analysed and reported on many of the therapies used in ME/CFS to improve fatigue and overall ill health [151]. However, few therapies have been subjected to randomised control trials (RCTs) using the newer more specific diagnostic criteria and few have had significant numbers enrolled in each arm. Importantly, while numerous supplements have been used for many years in ME/CFS, few have been suggested to specifically improve CD. However, L-carnitine was found to be significantly helpful in a retrospective before and after analysis of people with ME/CFS and was associated with improved serum serotonin levels [152]. This contrasts with an earlier study showing no benefit of the SSRI agent [153] in any aspect of ME/CFS symptomology, and recent animal work suggesting that ME/CFS may be due to increased serotonin within the brain [154]. Furthermore, aripiprazole in an open-label study improved brain fog as well as physical fatigue and unrefreshing sleep [155] suggesting that the modulation of dopamine pathways may be important in many of the symptoms in ME/CFS. Indeed, recent work suggests that dopamine is critical for neuronal plasticity and memory by its action on synaptic long term potentiation and projections to the hippocampus [156]. Interestingly, the EBV UTPase protein was reported to initiate neuroinflammation and modulate dopamine, serotonin, and tryptophan metabolism in a mouse model [86], thereby providing a link between viral infections and CD. Indeed HHV6 miRNA and EBV dUTPase have been found in several regions of the brain in ME/CFS but not in healthy controls [157]. The role of virally induced immune dysregulation and autoimmune mechanisms in ME/CFS formed the basis of using B cell depletion therapy and was initially successful in a randomised control trial (RCT) [158]. However, a later multicentre RCT with 151 patients failed to show benefit [159]. Although the precise reasons were unclear, it was speculated that the therapy only benefited those with definitive autoimmunity causing the ME/CFS symptoms. Immunoadsorption has also been utilised and found clinically helpful to remove harmful proteins including autonomic receptor auto-antibodies in ME/CFS [160,161]. Nevertheless, efforts to reduce neuroinflammation should be undertaken and will over time help CD and peripheral bodily symptoms [162]. In this respect, parallels with PASC have led to a small open-label study using the nebulised administration of a mixture of several agents with anti-oxidant action and aiming to reduce nuclear factor (NF) κB activation, with some benefit [163].

Non-restorative sleep with “prolonged latency, longer awake time after sleep onset, reduced sleep efficiency, decreased stage 2 sleep, more Stage 3, and longer rapid eye movement sleep latency” [164] is frequent in ME/CFS and contributes significantly to CD as well as impaired mood. Indeed, architecturally normal restorative sleep is critical in improving CD, restoring circadian rhythms, and helping learning, memory, and attention [165,166]. Moreover, the treatment of insomnia can improve chronic fatigue and improve cortisol recovery after stress [167]. In contrast, the prolonged disruption of normal sleep can contribute to neuroinflammation [168], norepinephrine-stimulated degeneration of the locus coeruleus, and hippocampal amyloid-β_42_ accumulation, with the potential to cause memory dysfunction and CD [169]. Unfortunately, the use of benzodiazepines, while sometimes reducing the time to fall asleep and helpful initially [170], frequently causes daytime drowsiness and can exacerbate CD in ME/CFS. Melatonin, however, is more suitable, can encourage the normalisation of sleep chronobiology, and has the ability to scavenge reactive oxygen species that are central to neuroinflammation [171]. Disappointingly, while melatonin is used by many with ME/CFS, it has not been systemically investigated for its potential benefit in this condition. However, combined with palmitoylethanolamide it has recently been reported to improve pain, sleep and overall the quality of life in patients with fibromyalgia [172] which is not infrequent in those with ME/CFS. Overall, it is surprising that more work has not been undertaken regarding the use of newer hypnotics and the dual orexin receptor antagonists in ME/CFS and whether any improvement in sleep duration and quality is associated with a simultaneous improvement in CD.

## 10. Future Directions

ME/CFS remains a complex, poorly understood, and inadequately investigated condition with numerous immune, CNS, endocrine, sleep, and biochemical abnormalities acting on a multidimensional biopsychosocial milieu. Although CD is frequent and highly disabling, it remains very difficult to evaluate and monitor. New methods involving computer-based assessment modules may allow more detailed analysis, especially regarding changes with time, mental activity, and treatment [173]. Additionally, identifying ME/CFS subtypes based on aetiology, the duration of disease, the mode of onset [174], disturbance of mood and sleep, ongoing viral reactivation symptoms, joint hypermobility syndrome with or without POTS [175,176], neurodiversity [173], and specific patterns of CNS and peripheral symptoms may allow for a more tailored treatment. Certainly, offering anti-inflammatory treatment to those ME/CFS patients with raised inflammatory markers needs consideration [177] and has provided benefits for the chronic fatigue in rheumatoid arthritis [178]. Nonetheless, the simultaneous treatment of each of the co-existing specific conditions would be needed to produce benefits while avoiding the potential for adverse drug interaction induced by polypharmacy.

The significant similarity of many of the symptoms of ME/CFS and (MS) [145] suggests that treatments used in the latter may provide benefit in the former. To this extent, glatiramer acetate, which is used in MS, has been found to bind abzymes (antibodies with enzyme activity) that can be seen in some patients with ME/CFS and can damage the myelin basic protein that covers nerves in the human body [179]. There is also mounting evidence of altered mitochondrial oxidation of the TCA cycle and pentose phosphate pathway intermediates, as well as of fatty acid β-oxidation and amino acid degradation [180]. Thus, efforts at restoring mitochondrial function using cocktails of factors that optimise mitochondrial energy production have been investigated, but unfortunately often as open-label studies using old diagnostic criteria or with insufficient power or attention to robust monitoring [181]. However, NADH, Coenzyme Q10, acetyl-L-carnitine, ubiquinol-10, guanidinoacetic acid, and glycerol supplementation were well investigated and found helpful [181]. This suggests that newer combinations of treatments should be considered in the future to fully optimise mitochondrial adenosine triphosphate (ATP) production. Importantly, patients with ME/CFS frequently have at least one other chronic pain condition such as fibromyalgia, chronic low back pain, irritable bowel syndrome, endometriosis, and chronic migraine [41]. Therefore, it is important that established treatments for these other conditions are also carefully utilised to reduce overall symptom burden. It is possible that this may reduce illness related stress leading to reduced neuroinflammation and hence chronic fatigue and CD.

Investigating CD in ME/CFS can be difficult when this is the major symptom. In this situation, the systemic inflammatory process may need consideration [139,140,141], especially if a recent SARS-CoV-2 infection has been documented. While cerebral spinal fluid analysis and meningeal/cerebral biopsy may be helpful, they represent significant hurdles and challenges. Presently, cerebral imaging using established TSPO ligands has presented difficulties in interpretation [73] and significant effort is often required by patients to endure the often prolonged tests. However, Zatcepin et al. [182] have used machine learning to determine the essential features of simplified TSPO-PET scanning in those with strokes. Combining this information with the targeted use of ^11^C-PBR28 MR-PET and 7T susceptibility-weighted imaging with machine learning, as utilised in a study of patients with MS [183], may present new opportunities.

## 11. Conclusions

Memory, attention, and cognition may be impaired because of peripheral inflammation and viral and bacterial infection. Peripherally synthesised pro-inflammatory cytokines or viral proteins interacting with specific CNS TLRs may be important in altering neuronal function. Locally released pro-inflammatory cytokines may also be important contributors to endothelial cell activation which may compromise cerebral perfusion and alter neurotransmitter synthesis. Brainstem dysfunction involving the ARAS may contribute to memory dysfunction by impairing attention. There is evidence that immune dysfunction resulting in autoimmunity or immune insufficiency may also be important in altering cognitive function in ME/CFS. Efforts to improve sleep and reduce even low levels of inflammation may have a beneficial effect on memory in patients with ME/CFS.

## Figures and Tables

**Figure 1 ijms-26-01896-f001:**
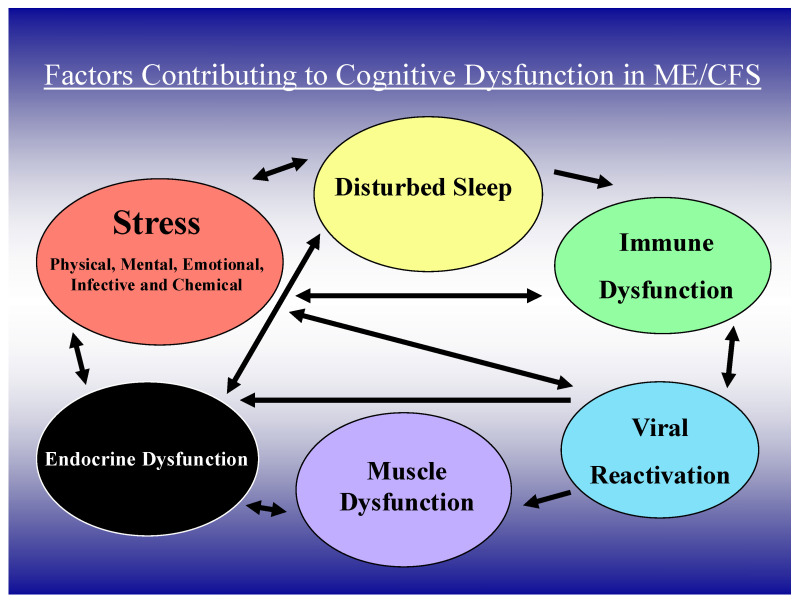
Clinical, immune, and viral factors important in cognitive dysfunction in ME/CFS.

**Table 1 ijms-26-01896-t001:** The important factors and the mechanisms by which they contribute to the cognitive dysfunction in ME/CFS.

Factor	Mechanism of Cognitive Dysfunction
Inflammatory cytokines	Neural cell dysfunction and disturbed neurotransmitter release, especially in the ARAS and hippocampus, and endothelial activation leading to impaired cerebral perfusion.
Brain-Derived Neurotrophic Factor	Diminished production by muscle tissue causes reduced neural cell proliferation.
Autonomic receptor auto-antibodies	Reduced blood pressure producing faintness and impaired cerebral perfusion.
Platelet activation	Micro-clot formation impairing cerebral perfusion and reducing oxygen and nutrient delivery to the brain.
Endocrine dysfunction	Impaired cortisol secretion affecting blood pressure and altered neural dendrite/synapse formation.
Viral infection/reactivation	Viral infection can directly affect neural cells and peripherally and centrally generated cytokines can directly disturb cognition. Viral factors can also decrease ATP production by mitochondria and encourage auto-antibody production.

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
