# Peer review of "Cognitive Dysfunction in Myalgic Encephalomyelitis/Chronic Fatigue Syndrome—Aetiology and Potential Treatments"

_ijms, 2025, doi:10.3390/ijms26051896_

Round 1

Reviewer 1 Report

Comments and Suggestions for Authors

The paper features good discussions of knowledge gaps in this area. However, it does not explicitly compare this review’s focus to other recent reviews on similar topics. For example, the manuscript does not reference the systematic review by Kim et al. (2020), which analyzed 56 randomized controlled trials (RCTs) and identified eight interventions with statistical significance but no definitive treatment for ME/CFS. Including these studies would strengthen the review by addressing pivotal research findings and ongoing controversies in ME/CFS management. While it suggests a need for stratification and subtype identification, it does not specify how this perspective uniquely differs from prior reviews.

The manuscript does not adequately address some key clinical trials in the area. For example, the manuscript does not explicitly address key studies such as the PACE trial (White et al., 2011), which evaluated adaptive pacing therapy (APT), cognitive behavior therapy (CBT), and graded exercise therapy (GET) but faced significant methodological criticisms, including broad inclusion criteria. The manuscript references the Fluge et al. (2019) trial indirectly by discussing B cell and antibody depletion strategies, noting their mixed or lack of success. However, it does not explicitly mention potential reasons why rituximab failed to produce clinical improvement and omits mention that it was associated with serious adverse events. Explicitly citing the Fluge trial and detailing its conclusions would provide greater context and strengthen the discussion of immune-targeted therapies in ME/CFS. 

The manuscript addresses differences across studies in the ME/CFS case definition. However, it does not adequately consider the nonspecific nature of ME/CFS symptoms or their overlap with other chronic conditions, except for Long COVID, which is noted as a potential trigger for ME/CFS.

The manuscript considers how future attempts might identify ME/CFS subtypes that respond differently to treatments but fails to mention a multidimensional biopsychosocial phenotype approach to this complex condition. While it discusses the need for stratification and subgroup identification to improve therapeutic targeting, it does not delve deeply into specific clustering methodologies. It notes the heterogeneity of cognitive and other symptoms across patients and the impact of different case definitions, suggesting subgrouping based on onset type (e.g., sudden infectious vs. gradual onset) or specific symptom profiles (e.g., cognitive impairments, viral reactivation) could help delineate phenotypes. These efforts aim to tailor interventions rather than relying on a one-size-fits-all approach.

The discussion of inflammation versus clinical subtypes for cognition-based evaluations in ME/CFS patients is somewhat superficial. A more detailed exploration of how inflammatory markers might correlate with specific subtypes could add value.

“There is therefore a need to delineate clinical subtypes for cognition-based evaluations in ME/CFS patients.” This statement could be expanded to specify how subtype delineation might inform clinical trials and therapeutic development.

The lack of well-designed clinical trials in the area is a major gap. This point deserves stronger emphasis in the manuscript, particularly in the context of heterogeneous populations and inconsistent case definitions.

The manuscript includes a discussion of the emotional and affective consequences of chronic fatigue syndrome and chronic inflammation as they relate to cognitive impairment, but this discussion could be more comprehensive. It mentions depression as a confound for neurocognitive impairment, citing studies like Raanes et al. that correlate psychological variables, sleep disturbances, and cytokine levels with cognitive deficits.

The section on TSPO could benefit from discussing newer radioligands and how they might address current limitations in interpreting TSPO findings in ME/CFS research.

The manuscript appropriately acknowledges that biomarkers and imaging findings often associated with ME/CFS, such as elevated pro-inflammatory cytokines and brain imaging abnormalities, are not specific to ME/CFS. It discusses how such findings are also observed in other neurological and systemic conditions. For example, cytokine elevations (e.g., IL-6, TNF-α) and imaging evidence of neuroinflammation or altered cerebral blood flow are common in other conditions involving systemic inflammation or immune dysregulation.

The manuscript states “Efforts at improving sleep and reducing even low levels of inflammation may have a beneficial effect on memory in patients with ME/CFS.” This point is well made, but it would benefit from elaboration on specific interventions that might address these factors, such as anti-inflammatory treatments or sleep therapies.

Although the manuscript touches on neurotransmitter changes in ME/CFS, the discussion of specific neurotransmitter changes, other than serotonin, is minimal. Expanding this discussion to include other neurotransmitters, such as dopamine and norepinephrine, could provide a more comprehensive perspective.

The manuscript lacks a clearly defined section on future work that highlights high-priority areas, such as clinical trials of specific agents or rigorous comparisons of biomarkers across neurological conditions like multiple sclerosis and Parkinson’s disease. Including such a section would significantly strengthen the discussion.

While the paper discusses autonomic dysfunction as a potential contributor to ME/CFS symptoms, including impaired cerebral perfusion and cognitive dysfunction, it does not explicitly mention or explore POTS and its connection to neurocognitive function in the context of ME/CFS. Adding this discussion would enhance the manuscript’s comprehensiveness.

Some clinical trials for ME/CFS have included participants with conditions causing CFS-like symptoms, while others have applied strict criteria to exclude such cases. The inclusion of heterogeneous populations complicates the interpretation of results, as these conditions might respond differently to interventions. For example, the PACE trial (2011) applied broad inclusion criteria based on the Oxford definition, which has been criticized for including participants with fatigue stemming from other conditions, such as depression or undiagnosed medical disorders. This has led to concerns about the generalizability of findings, particularly for cognitive behavioral therapy (CBT) and graded exercise therapy (GET), to patients with strictly defined ME/CFS under criteria like the Canadian Consensus Criteria or International Consensus Criteria.

The sentence “The second reviewed 33 studies published between 1988 and 2019, avoiding methodological divergences in tests and standards between studies, identifying significant impairments in visuo-spatial short-term and especially immediate memory, with impacts on working memory being consistent with a massive executive deficit in patients” requires clarification. What is meant by “avoiding” methodological divergences? If this refers to excluding studies with inconsistent methodologies, the language should explicitly state this.

The term “poly I-C” needs definition. It likely refers to polyinosinic-polycytidylic acid, a synthetic analog of double-stranded RNA used to simulate viral infection in research settings, but this should be explicitly clarified in the manuscript.

Author Response

We would again like to thank the 2 reviewers for diligent assessment of our report. Our responses to their comments are detailed in the attached file and recorded in red type. 

Reviewer 2 Report

Comments and Suggestions for Authors

This manuscript is a relatively thorough and scholarly review of what was a rare and debilitating disease that is now increasingly being recognized, though the aetiology and pathogenesis remain unclear. The published literature is well documented in the extensive list of references. The focus on cognitive/intellectual dysfunction is appropriate as it is one of the most disabling aspects of this disease, apart from the fatigue, motor disturbance and inertia.

One aspect of aetiology that the authors might consider adding in their discussion is chronic small-vessel cerebral vasculitis. A recently published study in children and adolescents, but which also can be extrapolated to adults, reviews the pathology of this autoimmune disorder (Noan AN, Dropol A, Tyrrell PN, et al. Toward a histological diagnosis of childhood small vessel CNS vasculitis. Pediatr Rheumatol 2024;22:111). In this context also, the authors might make a brief comment in their Discussion of the criteria and timing of cerebral biopsy in myalgic encephalomyelitis for more precise diagnosis that might indicate precision treatment in individual patients. Neuronal degeneration and synaptic density also can be demonstrated in tissue sections. Histopathology with immunocytochemical antibody labels compliment neuroimaging and genetic studies and provides another dimension not duplicated by other techniques; it can demonstrate inflammatory cells and markers in brain tissue that may differ from those in circulating blood samples. Pathology is not adequately addressed in this current manuscript.

References are appropriate in selection. Table 1 is a useful tabular summary of mechanisms contributing to cognitive dysfunction. Figure 1 is very simple and not essential.

Author Response

(The authors gave the same response as above.)

Reviewer 3 Report

Comments and Suggestions for Authors

I would like to thank the authors for their contribution and the time invested in this article.

In my opinion, the review has been carried out with clarity and good methodological quality.

I have only detected one small fault that is easily correctable, which is the absence of a description of the acronym EBV in the text. It is a virus that is referred to on several occasions and that appears for the first time in the article on line 227, but without an explanation of what the acronym corresponds to.

Although it is likely that people familiar with the subject of infections know perfectly well the meaning of these acronyms, I recommend the inclusion of a description of them in the text, to facilitate their understanding by the entire community.

Author Response

Comment 1:

I have only detected one small fault that is easily correctable, which is the absence of a description of the acronym EBV in the text. It is a virus that is referred to on several occasions and that appears for the first time in the article on line 227, but without an explanation of what the acronym corresponds to.

Although it is likely that people familiar with the subject of infections know perfectly well the meaning of these acronyms, I recommend the inclusion of a description of them in the text, to facilitate their understanding by the entire community.

Response 1: Corrected.

Round 2

Reviewer 1 Report

Comments and Suggestions for Authors

Major Outstanding Issues:
1. The manuscript still has numerous grammatical and clarity issues throughout. For example:
- Inconsistent tense usage (switching between present and past tense)
- Run-on sentences and awkward phrasing
- Missing articles and prepositions
- Non-standard scientific terminology
2. The discussion of clinical trials and interventions remains problematic:
- The PACE trial critique lacks sufficient methodological detail
- The discussion of rituximab trials needs clearer explanation of why they failed
- Treatment recommendations lack adequate evidence grading
- Cost-effectiveness data for interventions is not well integrated
3. The section on neurotransmitter changes is still minimal, particularly:
- Limited discussion of dopamine and norepinephrine systems
- Inadequate integration of serotonin findings with recent animal studies
- Missing mechanistic explanations for neurotransmitter alterations
4. The examination of inflammatory biomarkers remains incomplete:
- Specific inflammatory markers are not comprehensively detailed
- Relationship between inflammation and clinical subtypes is unclear
- Temporal relationships between inflammation and symptoms need better characterization
5. The manuscript still lacks:
- Clear delineation of ME/CFS subtypes
- Rigorous discussion of study limitations
- Detailed future research directions
- Comprehensive analysis of biopsychosocial factors

Comments on the Quality of English Language

As above